# Central Nervous System Involvement in Primary Sjögren’s Syndrome: Narrative Review of MRI Findings

**DOI:** 10.3390/diagnostics13010014

**Published:** 2022-12-21

**Authors:** László V. Módis, Zsófia Aradi, Ildikó Fanny Horváth, János Bencze, Tamás Papp, Miklós Emri, Ervin Berényi, Antal Bugán, Antónia Szántó

**Affiliations:** 1Department of Behavioural Sciences, Faculty of General Medicine, University of Debrecen, Móricz Zsigmond krt. 22, HU-4032 Debrecen, Hungary; 2Division of Clinical Immunology, Department of Internal Medicine, Faculty of General Medicine, University of Debrecen, Móricz Zsigmond krt. 22, HU-4032 Debrecen, Hungary; 3Division of Radiology and Imaging Science, Department of Medical Imaging, Faculty of General Medicine, University of Debrecen, Nagyerdei körút 98, HU-4032 Debrecen, Hungary; 4Division of Nuclear Medicine and Translational Imaging, Department of Medical Imaging, Faculty of General Medicine, University of Debrecen, Nagyerdei körút 98, HU-4032 Debrecen, Hungary

**Keywords:** Sjögren’s syndrome, magnetic resonance imaging, central nervous system, white matter hyperintensities

## Abstract

Central nervous system (CNS) involvement is one of the numerous extraglandular manifestations of primary Sjögren’s syndrome (pSS). Moreover, neurological complaints precede the sicca symptoms in 25–60% of the cases. We review the magnetic resonance imaging (MRI) lesions typical for pSS, involving the conventional examination, volumetric and morphometric studies, diffusion tensor imaging (DTI) and resting-state fMRI. The most common radiological lesions in pSS are white matter hyperintensities (WMH), scattered alterations hyperlucent on T2 and FLAIR sequences, typically located periventricularly and subcortically. Cortical atrophy and ventricular dilatation can also occur in pSS. Whilst these conditions are thought to be more common in pSS than healthy controls, DTI and resting-state fMRI alterations demonstrate evident microstructural changes in pSS. As pSS is often accompanied by cognitive symptoms, these MRI alterations are expectedly related to them. This relationship is not clearly delineated in conventional MRI studies, but DTI and resting-state fMRI examinations show more convincing correlations. In conclusion, the CNS manifestations of pSS do not follow a certain pattern. As the link between the MRI lesions and clinical manifestations is not well established, more studies involving larger populations should be performed to elucidate the correlations.

## 1. Introduction

Primary Sjögren’s Syndrome (pSS) is a chronic autoimmune exocrinopathy affecting 0.3–3.0% of the population [1]. Its main clinical hallmarks are the sicca symptoms (mucosal dryness manifesting mainly ocularly and orally) and extra-glandular symptoms, among which joint pain and chronic fatigue are the most important. Besides these symptoms, pSS may also manifest in the central nervous system (CNS), as noted in the original description of the disease by Sjögren himself in 1933 [2]. Although 8.5–70% of the patients with pSS suffer from nervous system abnormalities, which precede the sicca symptoms in 25–60% of the cases [3], the role of the CNS in pSS remains controversial. To establish the possible causes and consequences of CNS involvement in pSS, many studies have been conducted in recent decades. Even in the earliest ones, there is an effort to establish the connection between the clinical neuropsychological features and CNS imaging findings of patients.

The growing interest towards CNS involvement can be explained, among many others, by the fact that it may play a crucial role in the diagnosis of the disease. Neurological symptoms may be the first manifestations of the disease in at least 25% of the cases, preceding the diagnosis of pSS by 2 years, on average [4]. As it is, neurological manifestations, especially those detectable by MRI, may play an important role in the (early) diagnosis of pSS. On the other hand, however, it is easy to mix up early pSS with other conditions based on MRI findings. Therefore, MRI signs of pSS should be known and considered by physicians (mainly radiologists, immunologists, rheumatologists and neurologists) to provide the earliest diagnosis possible and hence to promote neuroprotection in pSS. This is a hard task since CNS alterations screened with MRI do not follow a distinct pattern, they are very varied, and sometimes it is difficult to match them to pSS, particularly in the absence of clinical signs of either neurological or internal nature.

In this review, we will summarize the previously published papers, which investigate the use of cerebral MRI in pSS, describe the detected alterations and give possible explanations for the etiopathogenesis, structure, clinical correlations and their possible impact on the disease course. This review contains a synopsis of the most common CNS MRI findings in pSS, the white matter hyperintensities, alongside results of volumetric and morphometric studies and fMRI studies. The etiopathogenesis of WMHs, the relationship between MRI-detected abnormalities and cognitive functions and other MRI findings will be discussed.

## 2. Methods

### 2.1. Study Design

For this narrative review, a comprehensive literature search was performed to identify all available papers related to the involvement of CNS in pSS examined with MRI. The search happened as described below. First, healthcare professionals and experts from the fields of internal medicine, radiology and behavioural medicine were recruited. The team set a search strategy determining the inclusion and exclusion criteria. This step was followed by the database search, identifying the publications accordant to our aim. Next, we selected the relevant publications from the search results by screening the abstracts and full texts when necessary. Finally, after data extraction, we summarized and interpreted the results. We focused, in particular, on the clinical and cognitive relevance of these alterations at the interpretation.

### 2.2. Search Strategy and Eligibility Criteria

The literature search happened through PubMed, combining the following key indexing terms: ‘primary Sjögren’s syndrome’, ‘central nervous system’ and ‘MRI’. The search resulted in 117 results. Reviews, case reports and consensus statements were excluded (case reports were used only in the discussion), as well as articles not written in English. Randomized controlled trials (RCTs), cohort studies (retrospective and prospective), non-randomized controlled trials and case-control studies investigating the CNS of pSS patients with MRI were included. Furthermore, studies without a detailed description of the MRI findings (e.g., studies with various methods only mentioning MRI) or animal studies were also excluded. In this manner, 31 articles fulfilled our inclusion criteria. Afterwards, we added 7 more relevant articles by reference tracking.

### 2.3. Data Extraction

Initially, all titles and abstracts found through the literature search were reviewed by independent authors from different clinical fields (behavioural medicine: LVM, internal medicine: AZs, IFH, radiology: JB, TP), and inclusion and exclusion criteria were verified. Then the full texts of the selected articles were analyzed with regard to the references to ensure that all relevant articles had been enrolled in the study. Finally, the following information was registered from the articles: first author, date of publication, title of the study, study design, sample size, presence and properties of WMHs and main findings.

## 3. Conventional MRI Studies

### 3.1. White Matter Hyperintensities (WMHs)

The most characteristic MRI finding in pSS is the white matter hyperintensities (WMHs). These are hyperintense foci on T2 weighted and FLAIR images typically located periventricularly and subcortically. These lesions can mimic the radiological properties of multiple sclerosis (MS). However, there are differences to be mentioned as well, as described below. The significance, role and etiopathogenesis of these lesions are still debated.

In a related study, four of twelve patients (33%) had such alterations, but neither the location nor the size could be correlated with any clinical CNS symptoms [5]. Moreover, regarding the immunological parameters, serum IgG, rheumatoid factor levels, and anticardiolipin antibodies do not correlate with WMHs properties [6]. There have also been investigations showing that the presence of WMHs on cerebral MRI between pSS patients and age and gender-matched controls do not differ statistically; therefore, they cannot be considered signs of CNS involvement but marks of physiological ageing [7]. However, this may also mean that the role of WMHs is underestimated because of the lack of proper diagnostical modalities [8]. More recent evidence suggests that there is an elevated number of WMHs in pSS patients [9,10]. Tzarouchi et al. compared pSS-affected cerebri to systemic sclerosis-affected ones and healthy controls and pointed out that while there were not any statistical differences between pSS and systemic sclerosis in terms of WMH numbers (independently from their size), the number of WMHs was significantly higher in pSS patients than in controls. Hence, WMHs as neuroradiological hallmarks of CNS involvement in pSS is still debated but based on recent research outcomes, it is more probable that they can be considered typical alterations of the CNS in pSS.

The clinical correlations of WMHs are also controversial. Coates et al. deny any significant clinical correlations of cerebral white matter lesions in pSS [6]. There is no connection between the number of pSS criteria fulfilled, extraglandular symptoms [5,7,11], fatigue [10,12] or depression [10] and WMHs. Patient age is found as the only clinical data significantly correlating with the number of WMHs [13]. More WMHs are to be found in younger patients [11], and pSS patients with WMHs tend to be younger than controls with WMHs [9]. Another study describes that whilst WMHs with demyelinating characteristics are more frequent with younger age and altered glomerular filtration rate, the total number of WMHs, however, is associated with higher age (contrary to the findings of Govoni et al.) as well as lower prevalence of leukopenia and anti-SSB (anti-La) autoantibodies, higher prevalence of hypertension, diabetes mellitus, metabolic syndrome and lower use of antimalarial drugs [14]. Disease duration is significantly associated with WMHs [9,15]. As for laboratory parameters, serum IgG, rheumatoid factor levels and anticardiolipin antibodies do not correlate with WMH properties as mentioned above [6]. The periventricular WMH number, however, is associated with increased platelet serotonin release in pSS patients with frequent episodic tension-type headaches without correlation to the actual EULAR Sjögren Syndrome Disease Activity Index (ESSDAI) and markers of immunoinflammation, and might be linked with chronic immunoinflammation of low-grade intensity and vasculitis in pSS [15]. Anti-AQP4 antibody, however, is associated with MRI lesions in the cerebrum, brainstem, optic nerve and posterior column of the cervical spinal cord [16]. WMHs can be observed in the spinal cord as well [17]. It is more typical though that a continuous hyperintense area can be observed on the T2 sequence in the spinal cord, mainly in the cervical part [18,19].

Results of resting-state fMRI and other imaging techniques also showed correlations with WMHs: the association of asymmetrical hypoperfusion verified by single-photon emission computerized tomography (SPECT) and subcortical WM lesions is statistically proven [20]. Furthermore, fMRI studies have detected a positive correlation in mean diffusivity (MD) in pSS patients in the anterior thalamic radiation, the corticospinal tract, the cingulum, the forceps minor and major, the inferior fronto-occipital fasciculus, the superior and inferior longitudinal fasciculus and the uncinate fasciculus [10]. Longitudinal studies and follow-up reports are available in a very limited number; the available data suggests that these lesions remain unchanged over time without progression [14,21], and patients with WMHs do not develop MS [6]. The evidence says that WMHs and cognitive functions also correlate, this is presented in detail in the chapter dedicated to cognitive functions.

The lack of a gold standard in the diagnosis of WMHs of different origins means that it is a complex, multidisciplinary task to evaluate the findings of each patient and to administer proper therapy.

### 3.2. Atrophy

Signs of atrophy are also typical in CNS imaging in pSS. Atrophy is not always present though: when directly measuring the cortical thickness of pSS patients, Segal et al. did not find any differences between pSS and healthy controls [22]. Cortical atrophy of 2 patients is reported based on CT scans [23]. The same condition was identified in 6 patients in an investigation involving 15 participants (31.25%). Focal alterations in the cortex were not seen [24]. The lack of focal atrophy was confirmed and extended to the WM: Lauvsnes et al. reported general atrophy both in the grey and white matter without any focal enhancement [25]. The mechanism of the atrophy is probably due to cortical hypoperfusion and the consequential lowered glucose metabolism that is present in pSS CNS. This hypoperfusion is established with SPECT studies [8,21,26] and possibly can be derived back to the vascular pathology mentioned before. An important unanswered question regarding this hypoperfusion is the problem of lateralization: the hypoperfused areas are more common on the left side, exhibiting asymmetry [8].

## 4. Volumetric and Morphometric Studies

The total volume of pSS patients’ brains is not significantly different compared to patients with migraine. However, ventricular volume was significantly higher in pSS with age covariation correction. This ventricular dilatation was associated with attention disturbance [27]. These alterations correlate with neuropsychological and psychiatric symptoms; thus, together with the previously mentioned SPECT studies would support an organic aetiology for these manifestations of pSS. Tzarouchi et al. performed the first voxel-based morphometric study in pSS. According to the results, the volume of both grey and white matter is reduced in the pSS group. In the grey matter (GM), the cortical regions were bilaterally affected, mainly in the occipital, parietal and frontal lobes; furthermore, the thalamus, caudate nucleus and cerebellar hemispheres were diminished. As for the WM, small areas with decreased volume could be observed throughout the brain, especially in the frontal and occipital lobes, cerebellum and corpus callosum (splenium, genu) [9]. On the contrary, a more recent study identified diffuse cerebral white matter loss in pSS patients but demonstrated the lack of GM or WM atrophy in specific areas of the brain [25]. In the search for an association between fatigue and brain volume, Hammonds et al. found no structural changes either in GM or WM that could be related to fatigue. These results apply to global volumes and individual brain regions [12].

## 5. DTI Studies

Diffusion tensor imaging (DTI) is an MRI-based technology based on the diffusion of water molecules, arranging the measure of the microstructural integrity of fibre tracts. Subtle tissue alterations that impact the integrity of the brain’s structural networks and interregional information transfer are visualized via DTI [28]. DTI revealed significant alterations between pSS patients with and without cognitive impairment and healthy controls in the inferior frontal WM. Lower fractional anisotropy (FA) and higher mean diffusivity (MD) reflecting the deterioration of physiological brain tissue microstructure were observed in the cognitively impaired pSS group [22]. Compared with control subjects, pSS patients turned out to have decreased FA bilaterally in the corticospinal tract, superior longitudinal fasciculus, anterior thalamic radiation, inferior fronto-occipital fasciculus, uncinate fasciculus and inferior longitudinal fasciculus. Voxelwise-based group comparison of MD, axial diffusivity (AD) and radial diffusivity (RD) between patients and healthy controls showed increased MD and RD and decreased AD in the CNS of pSS patients in an extensive, diffuse pattern involving most of the major WM tracts throughout the brain [29]. In another Greek study, pSS patients were divided into two groups: with and without depression. Patients with depression showed increased AD, RD and MD and decreased FA; those without depression showed decreased AD in major WM tracts (superior longitudinal fasciculus, inferior longitudinal fasciculus, corticospinal tract, anterior thalamic radiation, inferior fronto-occipital fasciculus, cingulum, uncinate fasciculus and forceps minor-major) [10]. Finally, in a paper published this year focusing on structural connectivity (SC), 12 connections were significantly different between pSS patients and healthy controls. Decreased SC in the frontal and parietal lobes and some parts of the temporal and occipital lobes were present in pSS patients. Furthermore, increased SC between the right caudate nucleus and the right median cingulate/paracingulate gyri was reported. The reduced SC between the left middle temporal gyrus and the left middle occipital gyrus was negatively associated with WMHs [30].

## 6. Resting-State fMRI Studies

Functional magnetic resonance imaging (fMRI) has been crucial to our current understanding of brain function. During task performance or in response to a stimulus, specific regions of the brain are activated and detected by fMRI. More recently, the modality has been developed for use at rest, termed resting-state fMRI or functional connectivity (FC) MR imaging. Resting-state fMRI investigates synchronous activations between regions that are distinct in space, appearing without a task or stimulus, to identify resting-state networks [31]. Thus, resting-state fMRI can provide valuable information about brain networks in pSS. The first fMRI study published on pSS was performed on 14 patients. The data were processed by regional homogeneity (ReHo) analysis. ReHo is the time consistency of the blood oxygenation level-dependent signal of local brain tissue [32]. The ReHo values were increased in the right cerebral hemisphere, left limbic lobe, right middle temporal gyrus and the inferior parietal lobe in pSS patients compared to controls. However, ReHo values were significantly decreased in the right lingual gyrus, left cuneiform lobe, left superior occipital gyrus, bilateral middle occipital gyrus and the bilateral fronto-parietal junction area [33]. Additionally, pSS patients were found to show decreased brain activation compared to controls in the sensorimotor network. No FC changes occurred when comparing patients with or without depression or fatigue and controls [10]. Another study focusing on the hippocampus revealed that hippocampal FC is decreased between the left hippocampus and the right inferior occipital and inferior temporal cortex, as well as between the right hippocampus and right inferior occipital and middle occipital GM, left middle occipital and left middle temporal GM. In addition, increased hippocampal FCs were detected between the left hippocampus and left putamen, as well as between the right hippocampus and right cerebellar posterior lobe. The FC of the right hippocampus and right inferior and middle occipital GM correlated positively with a visual reproduction score in this study, while WMH scores negatively correlated with the FC between the left hippocampus and right inferior occipital and inferior temporal GM [34]. The same group compared cerebral functional segregation in Sjögren’s syndrome (SS) with or without systemic lupus erythematosus (SLE). The measurement targeted spontaneous brain activity in SS-SLE and SS using the amplitude of low-frequency fluctuation (ALFF). ALFF differences occurred in the bilateral precuneus/posterior cingulate cortex, right parahippocampal gyrus/caudate/insula and left insula were found among the three groups. The right parahippocampal gyrus, right insula and left insula displayed decreased ALFF in both SS-SLE and SS compared to healthy controls, while the SS group showed decreased ALFF values in the right parahippocampal gyrus, right insula and left insula, and increased ALFF values in the bilateral posterior cingulate cortex [35].

## 7. Discussion

This study summarizes the MRI abnormalities detectable in pSS. The first chapter is devoted to the most common MRI lesions, the WMHs, of which the clinical relevance and significance have been debated from its first description. After reviewing the possible pathways of etiopathogenesis of these lesions, the previously compiled results will be discussed with special regard to their neuropsychological aspects.

### 7.1. Etiopathogenesis of WMHs

The etiopathogenesis and exact histological structure and, thus, the possible classification of the WMHs remain controversial. It is complicated by the fact that WMH appearance is rather heterogenous in terms of their number, size and location (Table 1). Many attempts have been made to find out the way of the emergence of these lesions. These theories included myelin pallor, dilatation of perivascular (Virchow–Robin) spaces, periventricular gliosis, arteriosclerosis and infarction [6]. It has recently been established that the two main pathways, which most likely interfere, are the vascular and demyelinating/inflammatory pathways. Microinfarcts or microhaemorrhages associated with a small-vessel cerebral vasculopathy have been demonstrated as pathogenic factors of WMHs. It was described in the case of similar foci observed in MS; however, in their case, there was no clinical sign of demyelination [36]. Abnormal cerebral blood flow was also detected in pSS: hypoperfusion in the parietal and temporal lobe was reported based on SPECT studies [21,26]. The hypoperfusion is asymmetrical and focal [8,20] and, due to the subsequent diminished glucose metabolism, is connected not only to WMHs but to atrophy too, which itself may be able to trigger WMH formation via Wallerian degeneration (see below). Cerebrovascular risk factors, such as hypertension, increase the incidence of WMHs, which also supports vascular origin [14]. Haemodynamic changes are also registered in pSS brains: higher mean pulsatility index and systolic–diastolic ratio was found in patients compared to controls, in correlation with anti-SSA autoantibodies. This suggests that the autoimmune response is involved in early cerebral haemodynamic dysfunctions. Additionally, functional impairment of the endothelium was established, which is possibly responsible for vasomotor dysfunction before any organic damage [13]. The importance of functional impairment prior to organic damage, and the role of the autoimmune response in CNS pathology is enhanced by a study examining somatosensory-evoked potentials (SEP) among 33 pSS patients without clinical features of CNS damage and normal head computed tomography scan. The relationship between SEP parameters and pSS disease duration, duration of arthralgia and presence of anti-SSA and SSB antibodies is also described in the study [37]. The same research group revealed abnormal brainstem auditory-evoked potentials (BAEP) investigating pSS patients without CNS involvement [38]. These bioelectrical activity dysfunctions may be a consequence of ongoing inflammatory and/or immunological processes, anticipating the detectable morphological changes. Vasculitis has been described in the brains of pSS patients [39,40,41]. Therefore, one may consider vasculitis as a pathogenic factor for WMH formation. It probably has low importance though: pSS is not a vasculitic disease despite the occurrence of small vessel vasculitis in some cases. The fact that pSS patients do not have more cerebral infarcts than healthy controls also provides evidence against vasculitis as a pathogenic factor [7,25]. In summary, vascular impairment is important but not the exclusive pathogenic mechanism in the aetiology of CNS involvement in pSS.

The other main pathway settles on inflammation and demyelination, with a high resemblance to MS [14,29,44,49]. Although it may be difficult to distinguish between MS-like and true-MS symptoms based on MRI, it is crucial since MRI may be the most important diagnostic tool to rely on in the differential diagnosis, especially at the beginning of the symptoms [50]. The differences in the etiopathogenesis of pSS and MS may also lead to different neuropathological and neuroradiological hallmarks: focal lesions are more common in pSS (Table 1), as well as lesions of vascular origin without clinical signs of demyelination. Furthermore, demyelinating lesions are not necessarily present in the brain of pSS patients. Clinical investigations can support the differential diagnosis; cerebrospinal fluid (CSF) analysis detects only one or two bands in pSS as opposed to the oligoclonal pattern found in MS [51]. Moreover, standard diagnostic tools for Sjögren’s syndrome (e.g., Schirmer test, sialometry, screening for autoantibodies (anti-SS-A, anti-SS-B) and salivary gland biopsy) can help differentiate between the two conditions.

Young age and lower prevalence of hypertension are significant variables for inflammatory/demyelinating lesions when it comes to the differentiation between vascular and inflammatory/demyelinating WMHs [14]. The presence of anti-aquaporin-4 (anti-AQP4) autoantibody in pSS is a predictive sign for demyelinating lesions; brain lesions fulfilling Barkhof’s criteria were found only in anti-AQP4 antibody-positive patients among 22 participants [16].

Similarly to anti-AQP4 antibodies, which are direct insults to the myelin, the autoimmune response itself can damage the CNS-promoting WMH formation. B-cell dysfunction is a key feature of pSS. The excessive B-cell activating factor (BAFF) production is a pathological mechanism in pSS [52]. Nonlymphoid cells, such as astrocytes, can express BAFF and trigger CNS manifestations.

Segal et al. revealed a connection between WM pathology and psychological features, resulting in another aspect of the etiopathogenesis of WMHs, namely stress-induced cytokine signalling [22]. Excessive or prolonged activation of the brain cytokine system leads to a dendritic loss in the prefrontal cortex and the apoptosis of astrocytes and oligodendrocytes [53]. Thus, it presumably damages the WM and strengthens the inflammatory response. There are a few more mechanisms that may have an impact on WMH formation. In WM structural changes of pSS, the role of Wallerian degeneration is raised [9,29]. Wallerian degeneration is a stereotypic procedure beginning within days after injury with the disintegration of axonal structures, followed by fragmentation-degradation of myelin caused by the infiltration of macrophages. Ultimately, it results in fibrosis and atrophy of the affected fibre tracts [29,54]. Atrophy is observed in cerebral MRI studies of pSS patients [24,36,44]; the neural loss might be a trigger for Wallerian degeneration. Subcortical localization of WMHs can support its role in the mechanism of WMH formation.

There are very few autopsy studies available to evaluate the histological background of WM changes in pSS. Histopathologic evaluation of the brains of pSS patients with CNS involvement showed small-vessel mononuclear inflammation and ischaemic-haemorrhagic vasculopathy. However, their relation to WMHs was not specified [55]. Neuropathological examination of a deceased 40-year-old woman with pSS demonstrated multifocal lesions in the cervical spinal cord and medulla in correspondence with previous MRI imaging. In addition, there was demyelination, spongy change and axonal swelling in the WM, but no remarkable vasculitic changes were observed in the CNS [56]. Furthermore, it is confirmed that CNS pathology in pSS may have non-vasculitic origin—contrary to early theories—such as in the case of non-vasculitic autoimmune inflammatory meningoencephalitis [57,58].

The etiopathogenesis of the WMHs in pSS-affected brains is still not delineated. Many factors take part in the emergence of these alterations. Hence their formation cannot be attributed to one single source. All the procedures mentioned above possibly play a role in the process of WMH formation.

### 7.2. MRI Findings and Cognitive Functions

The obvious neuropsychological and cognitive symptoms of pSS have drawn the attention of the scientific community in recent decades [59,60]. There is a wide range of cognitive involvement in pSS, from mild cognitive impairment (MCI) to severe dementia. There have been studies analysing cognitive functions and MRI findings together (Table 2). Their results, however, often contradict each other, which brings into question whether the cognitive impairments correlate with MRI abnormalities. On the other hand, there is concordance in the nature of cognitive manifestations in pSS. These are, for the most part, attention deficits, visual and verbal memory (working and long-term), psychomotor function and processing speed abnormalities (see references in Table 2). Possibly the immune system itself plays a role in these conditions; in patients with anti-SSA antibody positivity, higher cognitive function disorders can be observed [7,20,61,62]. The relationship between WMHs and cognitive deficits is an open question; many studies for and against have been written. More recent studies with relatively large sample sizes (Harboe et al. 2009, *n* = 68; Blanc et al. 2013, *n* = 25; Morreale et al. 2014, *n* = 81; Zhang et al. 2020, *n* = 38) claim an association between WMH load and cognitive functions. In addition, attention showed a statistically significant association with ventricular volume [27]. Segal et al. identified microstructural changes in the frontal WM in the brain of pSS patients, which correlated with MCI in pSS. A recent resting-state fMRI study found that visual reproduction score positively correlates with the FC between the right hippocampus and right inferior occipital and inferior temporal GM [34]. The correlation between MRI abnormalities and neuropsychometric tests is debated, but cognitive symptoms are unequivocally important and yet sometimes underestimate clinical symptoms of pSS.

The correlation of the cognitive symptoms, MRI findings and the atrophy typical for pSS described above all support the organic origin of mental health impairment in the disease. The left-sided hypoperfusion previously mentioned also supports this theory. The reason for the asymmetricity of the blood flow is an interesting and open question. Nevertheless, these procedures may cause detectable cognitive decline that can reach the threshold of clinical dementia [20,57]. Nation-wide, population-based studies also have confirmed that pSS patients have a higher risk for dementia [66,67]. To prevent and treat pSS-associated dementia, awareness of these conditions and early diagnosis should be a priority. Cognitive symptoms can be the first sign of pSS [68]; therefore, recognizing them may play a key role in the diagnosis of pSS in general. The studies published on this topic, however, often contradict each other and have not demonstrated a clear association between cognitive symptoms and MRI lesions. Once these correlations become more definitely established, assessing specific cognitive dysfunctions may be a promising screening tool to identify organic CNS alterations. When it comes to treatment, very little data is available about the effective treatment of neuropsychiatric involvement. Rituximab does not improve cognitive symptoms [69]. Orofacial pain conditions, however, can be treated with cognitive behavioural therapy (CBT) [70]. Thus, providing mental health support also increases the quality of life of patients by improving physical pain. A very recent study recommends that physicians, in their daily practice, consider the tight interplay between depression and anxiety on cognitive performance. This task may require interdisciplinary teamwork involving psychological or even psychiatric management [65].

### 7.3. Involvement of Brain Regions

A typical pattern or location of brain lesions has yet to be described in pSS. However, there are interesting findings regarding the involvement of certain territories of the brain. All the involved brain regions and WM tracts reviewed in this article cannot be discussed due to limited space. Hence we discuss findings about selected regions involved in pSS. The first such area is the visual cortex, where Tzarouchi et al. found atrophy [9], and Xing et al. detected abnormal brain activity with resting-state fMRI [33]. This finding can be explained by the association of pSS and neuromyelitis optica (NMO) [71], which may result in the degeneration of the entire circuitry [9]. Yan et al. describes abnormal spontaneous brain activities of limbic-cortical circuits in patients with dry eye disease. The significantly increased ReHo values in the left inferior occipital gyrus and decreased in the middle cingulum might play a role in the pathogenesis of dry eye disease [72]. Based on the clinical resemblance, perhaps a similar mechanism could be the reason for the visual cortex impairment in pSS.

The other affected region to discuss is the cerebellum. Govoni et al. mentions a >1cm hyperlucent area in the left cerebellar hemisphere of one patient [11]. The bilateral degeneration of cerebellar hemispheres with voxel-based morphometry is also described. WMHs can also appear in the cerebellum [14]. An autopsy showed high Ro52/TRIM21 expression in the Purkinje cells in the histological sections of the cerebellum in a patient with cerebellar degeneration. This may indicate that anti-Ro/SSA antibodies are the antineuronal antibodies involved in the cerebellar degeneration of patients with SS. This finding, besides the possible use of CSF as a biomarker in pSS, suggests that the autoimmune response linked to pSS causes cerebellar degeneration [73]. Although to our knowledge, no large studies have been conducted on the role of the cerebellum in pSS, many case reports are available about cerebellar involvement [74,75,76,77,78,79]. In conclusion, it is most likely that these are not sporadic cases; cerebellar impairment is possibly a rare but notable symptom of pSS.

## 8. Conclusions

The MRI alterations in pSS are very diverse and not organized as a well-recognizable clinical syndrome. The most characteristic abnormalities are the WMHs that are usually located periventricularly and subcortically. These WMHs may precede the sicca symptoms. Hence pSS must be considered in the differential diagnosis when they are identified. Although WMHs may be related to cognitive symptoms, the link between the MRI lesions and clinical manifestations is still controversial. More studies involving large sample sizes should be performed to elucidate the correlations. It is clear, however, that CNS pSS must be treated accounting for the individual symptoms of the patient by an interdisciplinary team from the earliest opportunity, involving internal medicine, neurology, clinical psychology and psychiatry.

## Figures and Tables

**Table 1 diagnostics-13-00014-t001:** Prevalence and properties of WMHs in patients with primary Sjögren’s syndrome. Abbreviations: N: neurological, P: psychiatric, NP/C: neuropsychological/cognitive, WMH: white matter hyperintensity, med: median, FS: Fazekas score.

	*n*	Mean Age (years)	Disease Duration (years)	Sex Ratio (f/m)	CNS Involvement (Clinical) *n* (%)	CNS Involvement Type	Presence of WMHs	Number/Range of WMHs	Size of WMHs (mm)	Typical Location of WMHs	WMHs Related to CNS Manifestation (+/−)	WMHs Higher in pSS Than Controls (+/−)	WMHs in Spinal Cord (+/−)
Alexander et al. 1988 [36] ^1^	38	52; 53 *	na	35/3 (19/3; 16/0 *)	16 (42.1%)	N;P;NP/C	13/38 (34.2%) (2/22 (9.1%); 12/16 (68.8%)*)	na	5–10	periventricular (9), subcortical (7)	+	na	na
Manthorpe et al. 1992 [5]	12	52 (med)	9 (med)	10/2	2 (16.67%)	P	4/12 (33.33%)	1–4	2–5	na	na	na	na
Pierot et al. 1993 [24]	15	60	na	12/3	1 (6.7%)	N	9/15 (60.00%)	10 or more (5), between 2 and 10 (2), less than 2 (2)	na	supratentorial white matter (9), basal ganglia (2)	-	na	na
Escudero et al. 1995 [42]	48	58.2	4.75	41/7	35 (72.9%)	N; NP/C	25 (51.3%)	na	5–10 (85–90%); >10 (10–15%)	white matter of centrum semiovale	+	+	na
Tajima et al. 1997 [43]	21	51.7	na	21/0	21(100%)	N	1/21 (4.8%)	na	na	periventricular	+	na	na
Al-Watban et al. 1998 [44]	6	na	na	6/0	6 (100%)	N	5/6 (83.3%)	na	na	white matter, spinal cord	+	na	+
Govoni et al. 1999 [11] ^1^	87	57.6; 58.8 *	5.5; 4.4 *	4/83 (3/66; 1/17 *)	7 (8%)	N;P;NP/C	6/7 (85.7%)	na	>10	cortico-medullar junction, periventricular areas	-	na	na
Coates et al. 1999 [6] ^2^	30	63	na	25/5	12 (40.0%); 6 (20.0%), 4 (13.3%), 1 (3.3%)	N;P	24/30 (80.0%)	0–36	<10	subcortical areas, deep white matter	na	+	na
Belin et al. 1999 [40]	14 SS;7 pSS	50.35	na	14/0	14/14 (100%)	NP/C	7/14	na	na	supra- and/or subtentorial	-	na	na
Lafitte et al. 2001 [45]	11	61.9	na	4/7	11 (100%)	N; NP/C	4/11 (36.4%)	na	na	periventricular areas	-	na	-
Mataro et al. 2003 [27]	15	55.7	4.46	15/0	na	N;P;NP/C	8/15 (53.3%)	na	na	periventricular areas (4); lobar areas (8); internal capsule (3); infratentorial (1)	+	-	na
Delalande et al. 2004 [17]	58	na	na	na	na	na	41/58 (70.0%)	na	na	na	+	na	+(49%)
Alhomoud et al. 2009 [46]	12	40.0	na	12/0	12/12 (100%)	N; NP/C	7/12 (58.3%)	na	na	subcortical areas, brainstem	na	na	+
Le Guern et al. 2010 [8]	10	40.2	7.19	10/0	8/10 (80.0%)	NP/C	8/10 (80.0%)	3–10 (2)	>2 (all)	frontoparietal subcortical regions	+	-	na
Massara et al. 2010 [21]	23	55.8	7	22/1	23/23 (100%)	N; NP/C	23/23 (100%)	na	na	periventricular and subcortical areas	-	na	na
Gono et al. 2011 [47]	10	na	na	na	10/10 (100%)	N	5/10 (50.0%)	na	na	na	-	na	na
Tzarouchi et al. 2012 [9]	53	63.07	10.05	52/1	0		38/53 (71.7%)	≥2 mm --> med: 1; range: 0–20; <2 mm --> med: 6; range: 0–24		associated with a reduced volume of GM in the cortex, deep GM, and cerebellum and WM loss in areas adjacent to regions of GM atrophy and in the corpus callosum	na	+	na
Akasbi et al. 2012 [14]	51	64.2	na	na	51/51 (100%)	N, NP/C	25/51 (49.0%)	isolated (<3) in 4 (16%) patients, multiple in 21 (84%) patients	na	corpus callosum (3), U fibres (3), cerebellum (3), pons (2), basal ganglia (1)	+	na	na
Yoshikawa et al. 2012 [20] ^3^	20	77.2	na	15/5	20/20 (100%)	NP/C	18/20 (90.0%)	single lesion (7), bilateral lesions (5), multiple lesions (5), confluent lesions (1)		subcortical areas	+	na	na
Sarac et al. 2013 [15]	22	58.5	8.95	18/4	22/22	N	ns	90 in total (0–75/patient; mean = 15.1)	<2 mm = 22; 2–5 mm = 47; >5 mm = 21	periventricular, subcortical, basal ganglia, cerebellum, mesencephalon, pons, infratentorial	+	+	na
Morreale et al. 2015 [13] ^4^	87	46.3	1.17	78/9	32/87 (36.8%); 28/87 (32,2%)	N; NP/C	21/87 (24,1%)	focal lesions (17), beginning confluence of lesions (4)	≥5	frontal, infratentorial and basal ganglia areas	-	-	na
Hammonds et al. 2017 [12]	64	58.1	6.9	56/9	na	na	52/64 (81.3%)	3 (median) (0–28)	na	na	-	na	na
Kurtulus et al. 2019 [48]	20	na	na	na	na	na	11/20	0–1 = 7; 2–3 = 3; 4–10 = 3; 10+ = 3	>3	cortical-subcortical and periventricular, localized in the frontal, parietal and occipital regions (most of the periventriculars in the parietal)	-	na	na
Andrianopoulou et al. 2020 [10]	29	61.6	10.9	29/0	11/29 (37.9%)	P	29/29 (100%)	10 (med) (1–120)	na	na	-	+	na
Zhang et al. 2020 [34]	38	50.73	3	38/0	na	NP/C	na	FS = 1	na	na	-	+	na
Zhang et al. 2022 [30]	41	49.98	3	41/0	na	NP/C	na	FS = 1	na	frontal (26), parietal (16), occipital (2), temporal (2) lobes	+	+	na

Footnotes: ^1^ patient groups with (marked with a star*) and without CNS involvement; ^2^ distribution of CNS involvement: 12 (40.00%) migraine headache; 6 (20.0%) depression, 4 (13.3%) paresthesia, 1 (3.3%) encephalitic illness; ^3^ lesions referred as subcortical lesions in the article, the term WMH/WMA is not used; ^4^ distribution of CNS involvement: 32 (36.8%) headache; 28 (32.2%) subclinical executive function disorder.

**Table 2 diagnostics-13-00014-t002:** MRI findings and cognitive functions in primary Sjögren’s syndrome. NP: neuropsychological, WMH: white matter hyperintensity, SS: Sjögren’s syndrome, pSS: primary Sjögren’s syndrome.

Reference	*n*	MRI Findings	Cognitive Involvement (*n*)	NP Diagnosis	Cognitive Symptoms	Correlation (MRI and Cognitive)
Alexander et al. 1988 [36]	38	WMH	16/38	NP testing	progressive dementia (4), attention and concentration defects (10), decreased verbal intelligence quotient (6), general orientation and information (3), new learning and recall (3), recall of visual stimuli (3), perseveration (2) and dysnomia (2)	+
Belin et al. 1999 [40]	7pSS; 14SS	WMH	7/7 (14/14)	NP testing	face naming (language) (3); frontal lobe functions: mild (2), moderate (5); constructional praxis (2); face recognition (1); verbal working memory (2); verbal (1) and visuospatial (2) long-term memory; incidental memory (5)	-
Lafitte et al. 2001 [45]	36	various, mainly WMH	8/36	clinical examination, NP testing	subcortical or corticosubcortical dysfunction	-
Mataro et al. 2003 [27]	15	WMH and larger ventricular volume	7/15	NP testing	frontal lobe functions, memory	+
Harboe et al. 2009 [63]	68	WMH	34/68 (16 mild, 14 moderate, 4 severe)	NP testing	examined: attention, complex attention, memory, visual-spatialprocessing, language, reasoning/problem solving, psychomotor speed, motor function	+
LeGuern et al. 2009 [8]	10	WMH (2/10)	8/10	NP testing	executive and visuospatial disorders	-
Segal et al. 2010 [22]	19	frontal region WM microstructure alterations (DTI)	8/19	clinical examination, NP testing	psychomotor speed and sustained attention, working memory, attention	+
Yoshikawa et al. 2012 [20]	20	subcortical lesions (18/20)	20/20 (all were memory clinic patients)	clinical examination, NP testing	mild cognitive impairment (7), dementia (13)	-
Blanc et al. 2013 [64]	25	WMH	15/25	NP testing	mild cognitive impairment (speed of information processing, attention, immediate and long-term memory, executive functions) (10), dementia (5)	+
Morreale et al. 2014 [61]	81	WMH (infarctions, MS-like foci)	36/81	NP testing	subcortical frontal executive functions and verbal memory	+
Kurtulus et al. 2019 [48]	22	WMH	na	NP testing	delayed recall, multiple choice (MoCa)	-
Zhang et al. 2020 [34]	38	altered hippocampal functional connectivity	na	NP testing	psychomotor function, attention, processing speed, visual memory	+(visual reproduction)
Zhang et el. 2022 [30]	41		na	NP testing	impaired psychomotor function, processing speed, visual memory	+
Goulabchand et al. 2022 [65]	32	WMH, hippocampal atrophy	32/32	clinical examination, NP testing	mild cognitive impairment	-

## Data Availability

Not applicable.

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
