# Peer review of "Central Nervous System Involvement in Primary Sjögren’s Syndrome: Narrative Review of MRI Findings"

_diagnostics, 2022, doi:10.3390/diagnostics13010014_

Round 1

Reviewer 1 Report

The group of investigating patients is not appropriate defined; age, duration of disease and presence of certain type of specific antibodies???

The distinction to Multiple sclerosis is not enough stressed.

Reviewer 2 Report

The authors did a systematic review on brain MRI findings in primary Sjögren's syndrome (pSS). The topic is interesting and of much interest because neurological symptoms are a very common manifestation of pSS, and the idea that autoimmune diseases of a particular system only affect it has long been abandoned.

However the paper is fairly written and provides a good analysis of the results it has some inaccuracies that should be corrected before publication:

1. what was the methodology of the work? what were the rules for searching for articles? The authors should provide an analysis of qualified/disqualified articles, e.g., using a flow-chart or descriptively. We do not know what keywords were used to find relevant literature. 

2. in the discussion, it is worth discussing some studies suggesting brain bioelectrical dysfunction in pSS patients without nervous system damage measured by evoked potentials based on:

https://pubmed.ncbi.nlm.nih.gov/29850640/

https://pubmed.ncbi.nlm.nih.gov/32809145/

these studies may confirm the discussed functional impairment in pSS
